# Maternal vaccination and protective immunity against Zika virus vertical transmission

Chao Shan [1,2,13], Xuping Xie[1,13], Huanle Luo[3], Antonio E. Muruato[3], Yang Liu[1], Maki Wakamiya[1], Jun-Ho La[4], Jin Mo Chung[4], Scott C. Weaver[3,5,6,7,8,9,10], Tian Wang[3,7,11] & Pei-Yong Shi[1,5,6,7,8,11,12]*

An important goal of the Zika virus (ZIKV) vaccine is to prevent a congenital syndrome in fetuses of pregnant women, but studies directly evaluating maternal vaccination for ZIKV are lacking. Here we report maternal vaccination using a live-attenuated ZIKV vaccine (3′UTR-Δ10-LAV) in a pregnant mouse model. Maternal immunization with 3′UTR-Δ10-LAV does not cause any adverse effects on pregnancy, fetal development, or offspring behavior. One maternal immunization fully protects dams against ZIKV infection and in utero transmission. Although neutralizing antibody alone is sufficient to prevent in utero transmission, a higher neutralizing titer is required to protect pregnant mice against in utero transmission than that required to protect non-pregnant mice against viral infection. The immunized dams transfer maternal antibodies to pups, which protect neonates against ZIKV infection. Notably, pregnancy weakens maternal T cell response to 3′UTR-Δ10-LAV vaccination. Our results suggest that, besides vaccinating non-pregnant individuals, 3′UTR-Δ10-LAV may also be considered for maternal vaccination.

[1] Department of Biochemistry & Molecular Biology, University of Texas Medical Branch, Galveston, Texas, USA. [2] Wuhan National Biosafety Laboratory, Mega-Science Center for Bio-Safety Research, Chinese Academy of Sciences, Wuhan, Hubei, China. [3] Department of Microbiology & Immunology, University of Texas Medical Branch, Galveston, Texas, USA. [4] Department of Neuroscience, Cell Biology, & Anatomy, University of Texas Medical Branch, Galveston, Texas, USA. [5] Institute for Human Infections & Immunity, University of Texas Medical Branch, Galveston, Texas, USA. [6] Institute for Translational Sciences, University of Texas Medical Branch, Galveston, Texas, USA. [7] Sealy Institute for Vaccine Sciences, University of Texas Medical Branch, Galveston, Texas, USA. [8] Sealy Center for Structural Biology & Molecular Biophysics, University of Texas Medical Branch, Galveston, Texas, USA. [9] Center for Biodefense & Emerging Infectious Diseases, University of Texas Medical Branch, Galveston, Texas, USA. [10] World Reference Center for Emerging Viruses and Arboviruses, University of Texas Medical Branch, Galveston, Texas, USA. [11] Department of Pathology, University of Texas Medical Branch, Galveston, Texas, USA. [12] Department of Phamarcology & Toxicology, University of Texas Medical Branch, Galveston, Texas, USA. [13] These authors contributed equally: Chao Shan, Xuping Xie. *email: peshi@utmb.edu

Zika virus (ZIKV) is a mosquito-borne flavivirus that has recently emerged and caused major epidemics[1]. The most devastating disease of ZIKV infection is congenital Zika syndrome, including microcephaly, congenital malformations, and demise fetuses from infected pregnant women[2,3]. In adults, ZIKV infection is associated with Guillain-Barré syndrome (GBS), an autoimmune disease that can lead to paralysis[4,5]. One out of seven infants born to mothers with laboratory-confirmed ZIKV infection during pregnancy has birth defects, most likely due to in utero transmission[6]. Its teratogenic potential and explosive epidemics led the World Health Organization (WHO) to declare ZIKV as a Public Health Emergency of International Concern in February 2016. Since then, various ZIKV vaccine platforms have been developed, with the goal to control future epidemics and to prevent congenital syndrome. These vaccine platforms include inactivated, subunit, and live-attenuated vaccines[7], some of which have already entered clinical phase I/II trials[8–10]. We previously developed a live-attenuated ZIKV vaccine with a 10-nucleotide deletion in the 3′UTR of viral genome (3′UTR-Δ10-LAV). A single-dose immunization of 3′UTR-Δ10-LAV protected against ZIKV infection and in utero transmission in mice and non-human primates[11,12]. This vaccine candidate has an excellent safety profile, as evidenced by >1000-fold lower neurovirulence than the two licensed, live-attenuated flavivirus vaccines (yellow fever 17D and Japanese encephalitis SA 14-14-2)[13].

Despite the above promising progress, a number of important questions about ZIKV vaccine research remain to be addressed. Could maternal vaccination (i.e., vaccination during pregnancy) protect against viral infection and in utero transmission? Does pregnancy affect immune responses to ZIKV vaccination? Does maternal immunity derived from vaccination during pregnancy protect newborns against infection? Do pregnant and non-pregnant individuals require different levels of immunity to prevent ZIKV infections? Is neutralizing antibody alone sufficient to prevent in utero transmission? The answers to these questions are critical to guide the clinical development and regulatory approval of ZIKV vaccines.

In this study, we have addressed the above questions by testing 3′UTR-Δ10-LAV in the A129 (interferon α/β receptor-deficient) pregnancy mouse model[14,15]. Our results show that maternal vaccination of 3′UTR-Δ10-LAV is safe for pregnant mice and fetuses. A single-dose maternal immunization fully protects dams against ZIKV infection and in utero transmission. Neutralizing antibody alone is enough for protection against in utero transmission. However, a higher neutralizing titer is required to prevent in utero transmission in pregnant mice than that required to prevent viral infection in non-pregnant mice. In addition, pregnancy weakens T cell response to 3′UTR-Δ10-LAV immunization. Furthermore, maternal neutralizing antibodies are transferred to offspring to prevent ZIKV infection.

## Results

### Rapid immunity and long-lasting neutralizing antibodies.
We chose 3′UTR-Δ10-LAV in the current study because it elicits rapid and long-lasting protective immunity after one immunization. A single immunization of ten-week-old A129 mice with $10^3$ FFU of 3′UTR-Δ10-LAV elicits neutralizing antibody titers ($NT_{50}$, defined as a sera dilution that inhibits 50% viral infection, data represents $1/NT_{50} \pm$ standard deviations) of $1/1300 \pm 310$, $1/6800 \pm 900$, and $1/13,900 \pm 3000$ on days 6, 10, and 14 post-immunization, respectively (Supplementary Fig. 1). The $NT_{50}$ values remain above 1/20,000 at week 55 post-immunization (Supplementary Fig. 2). Given the short gestation period of 20 days in mice, a rapid development of immunity after 3′UTR-Δ10-LAV immunization is required to study maternal

vaccination in a mouse pregnancy model. Other vaccine platforms that require multiple doses (e.g., inactivated vaccine or DNA subunit vaccine) are not suitable for studying maternal vaccination in mice because they cannot elicit protective immunity within the short gestation period in the mouse model. Thus, we chose 3′UTR-Δ10-LAV to evaluate maternal vaccination in a pregnant A129 mouse model.

### Safety analysis of 3′UTR-Δ10-LAV for maternal vaccination.
Safety is paramount for vaccine development. We performed three experiments to test the safety of 3′UTR-Δ10-LAV in A129 pregnant mice. The first experiment examined potential adverse effects when mice were immunized during early pregnancy. Ten- to twelve-week-old A129 female mice were mated and closely observed for vaginal plugs [that define embryonic day 0.5 (E0.5)]. At E0.5, the pregnant mice were subcutaneously infected with $10^5$ FFU wild-type (WT) ZIKV or 3′UTR-Δ10-LAV. At E18.5, the animals were sacrificed and analyzed for fetal development, viral loads, and neutralizing antibody titers (Fig. 1a). In the WT virus group, 54% of the fetuses were resorbed (Fig. 1b) and the weights of non-resorbed fetuses were significantly lower than those from the PBS-challenged placebo group (Fig. 1c). In contrast, 3′UTR-Δ10-LAV infection did not affect fetal resorption or fetal body weight (Fig. 1b–d). No infectious virus was detected in maternal spleen (Fig. 1e), brain (Fig. 1f), placenta (Fig. 1g), or fetal head (Fig. 1h) from both WT- and 3′UTR-Δ10-LAV-infected groups at E18.5, suggesting that viral infections had been cleared by this time point. To increase the detection sensitivity, we also measured viral RNA loads in placenta and fetal head using quantitative RT-PCR (qRT-PCR). No viral RNA was detected in placenta (Fig. 1i) or fetal head (Fig. 1j) from the 3′UTR-Δ10-LAV-infected mice at E18.5; however, the WT virus-infected animals developed viral RNA in every placenta (100%), but none in fetal head from the non-absorbed fetus (0%; Fig. 1i–j). To further demonstrate that 3′UTR-Δ10-LAV does not infect fetuses at earlier gestation timepoints, we performed qRT-PCR on the placenta and fetus/fetal head harvested at E10.5 and E14.5 (Supplementary Fig. 3a); the results showed that 34.6% (9/26) and 7.4% (2/27) of the placenta developed viral RNA, respectively; no viral RNA was detected from any of the fetus/fetal head (Supplementary Fig. 3b, c). At E18.5, robust neutralizing antibody titers were detected in dams ($\sim 1/15,300 \pm 7000$; Fig. 1k) and fetuses ($\sim 1/3700 \pm 1800$; Fig. 1l) from both the WT and 3′UTR-Δ10-LAV groups. These results suggest that maternal vaccination of 3′UTR-Δ10-LAV does not cause fetal infection, and that maternal-to-fetal antibody transfer may contribute to the neutralizing activity in fetuses.

The second safety experiment examined potential adverse effects when mice were immunized at E10.5 (Supplementary Fig. 4a). We chose E10.5 to vaccinate the mice after placenta has been formed at E9.5 (Supplementary Fig. 5). Infection of pregnant mice at this time point with WT or 3′UTR-Δ10-LAV did not increase fetal resorption (Supplementary Fig. 4b). However, WT virus, but not 3′UTR-Δ10-LAV, caused significant fetal weight loss (Supplementary Fig. 4c). Corroboratively, infectious virus was detected in maternal spleen (50%), brain (100%), and placenta (100%) from the WT ZIKV group; in contrast, no infectious virus was found in any organs from the 3′UTR-Δ10-LAV group (Supplementary Fig. 4d, f). No infectious virus was detected in the fetal brains from WT and 3′UTR-Δ10-LAV groups under this experimental condition (Supplementary Fig. 4g). Corroboratively, qRT-PCR showed that the WT virus-infected animals developed viral RNA in placenta (100%; Supplementary Fig. 4h) and fetal head (100%; Supplementary Fig. 4i). Although the 3′UTR-Δ10-LAV-infected mice also developed viral RNA in placenta, the viral RNA load was 673-fold lower than that in the WT

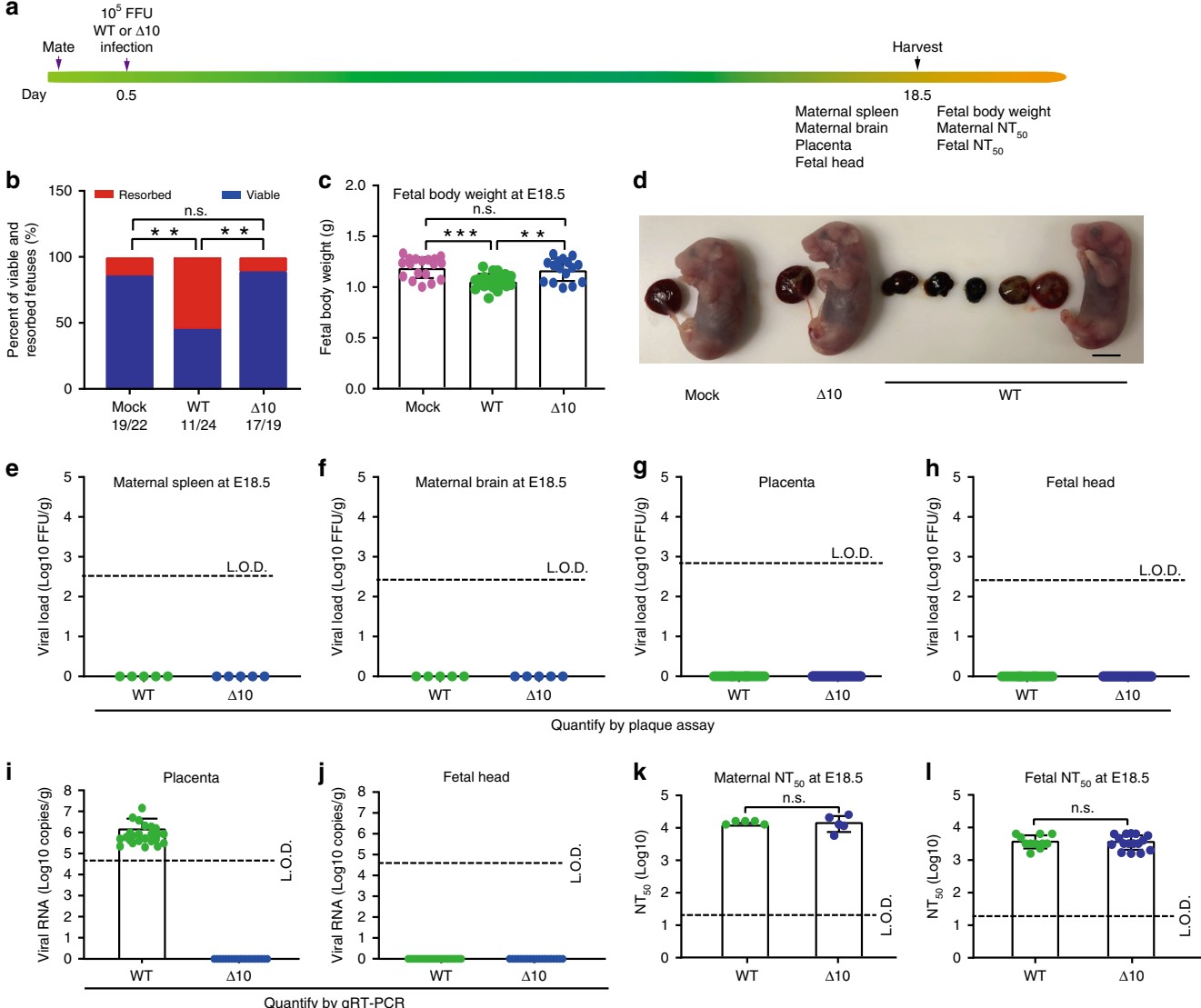

**Fig. 1 The safety of 3′UTR-Δ10-LAV in pregnant A129 mice when vaccinated at E0.5. a** Experimental scheme. Ten- to twelve-week-old pregnant mice were subcutaneously infected with $10^5$ FFU WT ZIKV ($n = 5$) and 3′UTR-Δ10-LAV (Δ10; $n = 5$) at E0.5. Maternal and fetal tissues were harvested and analyzed at E18.5. **b** The percentage of fetuses that were resorbed during pregnancy ($n = 22$ for mock; $n = 24$ for WT ZIKV; $n = 19$ for Δ10 vaccine; chi-square test [**$p < 0.01$]). The numbers of normal fetuses and total fetuses are presented below each group. **c** Fetal body weight at E18.5 for placebo, WT ZIKV, and Δ10 groups. Asterisks indicate significant differences as analyzed by one-way ANOVA. **$p < 0.01$, ***$p < 0.001$, $p > 0.5$ non-significant (n.s.). **d** Representative images of fetuses collected at E18.5 from placebo, Δ10, and WT ZIKV groups. Scale bar, 5 mm. Viral loads quantified by plaque assay at E18.5 are presented for maternal spleen (**e**), brain (**f**), placenta (**g**), and fetal head (**h**). Viral RNA measured by quantitative RT-PCR (qRT-PCR) at E18.5 are presented for placenta (**i**) and fetal head (**j**). **k** Maternal neutralizing antibody titers from WT ZIKV- and Δ10-infected pregnant mice at E18.5 with duplicate technical replicates. n.s. non-significant (Mann–Whitney test). **l** Neutralizing antibody titers in fetuses from WT ZIKV- and Δ10-infected dams at E18.5 with duplicate technical replicates. L.O.D. limit of detection, n.s. non-significant (Mann–Whitney test). Error bars represent standard deviations. Source data are provided as a Source Data file.

virus-infected placenta (Supplementary Fig. 4h). Furthermore, no viral RNA was detected in the fetal head from the 3′UTR-Δ10-LAV-infected animals (Supplementary Fig. 4i). At E18.5 or E21, comparable neutralizing antibody titers were detected in dams from the WT and 3′UTR-Δ10-LAV groups (Supplementary Fig. 4j). In contrast, no neutralizing activities were detected from the fetuses collected at E18.5, but low neutralizing antibody levels (~1/130) were found in the pups born at E21 (Supplementary Fig. 4k), suggesting that vaccination at E10.5 may not allow sufficient time to develop maternal IgG antibodies that are transferrable to fetuses[16].

The third safety experiment examined the effect of maternal vaccination on pups' development and behavior. Behavior tests are believed to provide a more sensitive readout for potential adverse effects than the parameters measured in the two preceding safety experiments. Pups delivered from maternally vaccinated dams (subcutaneously immunized with 3′UTR-Δ10-LAV at E0.5) were subjected to three behavior tests (including rotarod, grip strength, and tail pressure tests) when they reached 6-week-old of age (Supplementary Fig. 6a). Maternal vaccination did not change pup's body weight, body length, or behavior scores (Supplementary Fig. 6b–f).

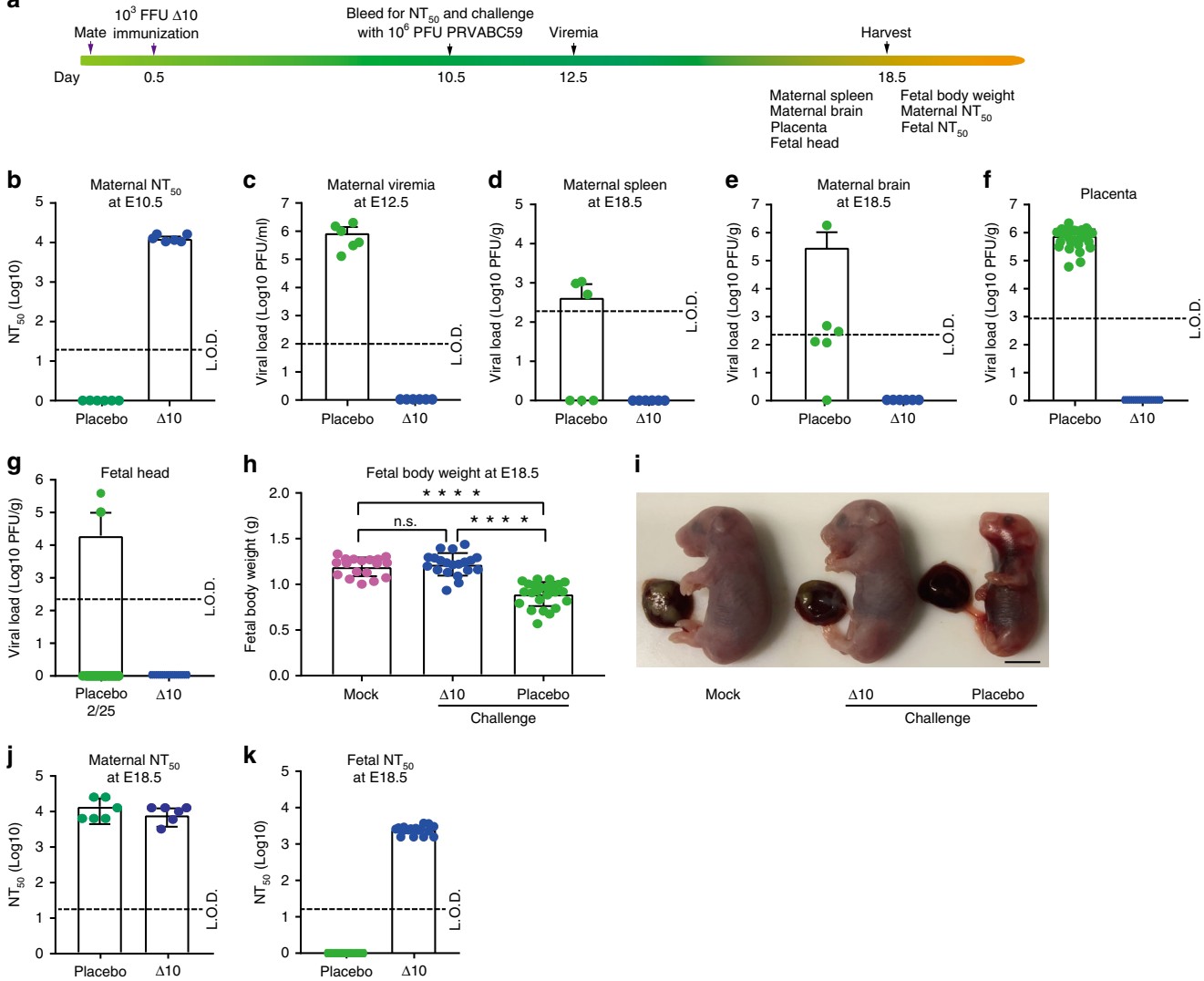

**Fig. 2 The efficacy of 3′UTR-Δ10-LAV in pregnant A129 mice when immunized at E0.5. a** Experimental scheme. At E0.5, ten- to twelve-week-old pregnant mice were subcutaneously immunized with $10^3$ FFU 3′UTR-Δ10-LAV (Δ10, $n = 6$) or PBS placebo ($n = 6$). At E10.5, the pregnant mice were bled for $NT_{50}$ measurement and subcutaneously challenged with $10^6$ PFU of ZIKV PRVABC59. Viremia was quantified by at E12.5. Maternal and fetal tissues were harvested for analysis at E18.5. **b** Neutralizing antibody titers from PBS- or Δ10-immunized pregnant mice at E10.5. **c** Viremia at E12.5. Viral loads at E18.5 are presented for maternal spleen (**d**), brain (**e**), placenta (**f**), and fetal head (**g**). (**h**) Fetal body weights at E18.5 from three experimental groups: non-immunized, un-challenge (mock; *left panel*); Δ10-vaccinated, challenged (*middle panel*); and PBS-immunized, challenged (*right panel*). Asterisks indicate significant differences (one-way ANOVA). ****$p < 0.0001$; non-significant (n.s) $p > 0.5$. Because the experiments in Figs. 1 and 2 were performed at the same time, the mock group in (**h**) is the same as the mock group in Fig. 1c; this reduced the number of animal use. (**i**) Representative images of fetuses collected at E18.5 from the three groups in (**h**). Scale bar, 5 mm. (**j**) Maternal neutralizing antibody titers from pregnant mice at E18.5. (**k**) Fetal neutralizing antibody titers from fetuses at E18.5. Duplicate technical replicates were performed for (**j**) and (**k**). Error bars represent standard deviations. Source data are provided as a Source Data file.

Taken together, the above safety results showed that maternal vaccination with 3′UTR-Δ10-LAV did not yield any detectable adverse effects on pregnancy, fetal development, or offspring behavior. We previously showed that a single immunization with 10 FFU of 3′UTR-Δ10-LAV elicited sterilizing immunity (defined by no increase of neutralizing antibody titers after WT ZIKV challenge) in A129 mice[12]. Thus, the dose of $10^5$ FFU that we used in the above safety experiments is 10,000-fold higher than the 10-FFU sterilizing dose, underscoring the excellent safety profile of 3′UTR-Δ10-LAV.

**Efficacy of 3′UTR-Δ10-LAV for maternal vaccination.** We tested the efficacy of 3′UTR-Δ10-LAV in maternal vaccination in A129 mice. Ten- to twelve-week-old pregnant A129 mice were

subcutaneously immunized with $10^3$ FFU 3′UTR-Δ10-LAV at E0.5 (Fig. 2a). High neutralizing antibody titers of ~1/13,200 ± 1700 were detected at E10.5 (Fig. 2b). After subcutaneously challenged with $10^6$ PFU Puerto Rico ZIKV strain (PRVABC59) at E10.5, the PBS placebo group developed $8.9 \times 10^5$ PFU/ml viremia at E12.5, whereas no viremia was detected from the 3′UTR-Δ10-LAV group (Fig. 2c). At E18.5, infectious virus was recovered from maternal spleen (50%), brain (83%), placenta (100%), and fetal brain (8%) from the placebo group; whereas no virus was detected in any organs from the 3′UTR-Δ10-LAV-vaccinated animals (Fig. 2d-g). In addition, 3′UTR-Δ10-LAV vaccination also prevented fetal weight loss (Fig. 2h-i). Compared with the neutralizing antibody titers detected at E10.5 (Fig. 2b), ZIKV challenge did not increase the maternal neutralizing titers

at E18.5 (Fig. 2j), indicating that the pregnant mice had achieved sterilizing immunity after vaccination. In addition, high maternal neutralizing antibody titers ($1/2,600 \pm 600$) were detected in the fetuses from the vaccinated dams (Fig. 2k).

Besides vaccination at E0.5, we also tested the efficacy of vaccination at E4.5 (Supplementary Fig. 7). In the mouse pregnancy model, E4.5 is the latest time that could be rationalized for maternal vaccination because (i) six days are needed to allow 3′UTR-Δ10-LAV to elicit protective immunity (Supplementary Fig. 1) and (ii) E10.5 is an optimal infection time to achieve analyzable fetal viral loads and developmental defects[15]. Similar to vaccination at E0.5 (Fig. 2), immunization at E4.5 elicited robust neutralizing antibody titers ($\sim 1/5325 \pm 2148$; Supplementary Fig. 7b) that fully prevented maternal and fetal infections as well as fetal weight loss (Supplementary Fig. 7c-h). Challenge with ZIKV PRVABC59 did not significantly boost maternal neutralizing antibody titers in the 3′UTR-Δ10-LAV-vaccinated dams (Compare Supplementary Fig. 7b, i). Maternal antibody titers of $\sim 1/980 \pm 528$ were detected in the fetuses collected at E18.5 (Supplementary Fig. 7j). Altogether, these results demonstrate that a single maternal vaccination of $10^3$ FFU 3′UTR-Δ10-LAV prevents in utero transmission during pregnancy. Under both vaccination schemes at E0.5 and E4.5, maternal neutralizing antibodies ($1/2600 \pm 600$ and $1/980 \pm 530$, respectively) were transferred to pups (Fig. 2k and Supplementary Fig. 7j).

**The effect of maternal vaccination on offspring protection.** It is important to characterize the effect of maternal vaccination on offspring protection against ZIKV infection. We performed three experiments to address this question (Fig. 3a). The first experiment examined the protection of maternally transferred immunity against ZIKV infection in pups. Dams subcutaneously immunized with $10^3$ FFU 3′UTR-Δ10-LAV at E0.5 developed comparable neutralizing antibody titers of $\sim 1/10,000$ at E10.5 and E21 (Fig. 3b). One-day-old pups born to the vaccinated dams showed an average neutralizing titer of $\sim 1/3000 \pm 1100$ (Fig. 3c). The milk recovered from the pups' milk spot exhibited an average neutralizing titer of $\sim 1/2500 \pm 800$ (Fig. 3c). These one-day-old pups survived a subcutaneous challenge of 100 PFU PRVABC59 (Fig. 3d), whereas naïve pups succumbed to challenges as low as 1 PFU ZIKV (Supplementary Fig. 8). The results demonstrate that maternal antibodies (transferred to pups through placenta and milk) can protect neonates against ZIKV infection.

The second experiment examined the longevity of maternal immunity in offspring. The neutralizing antibody titers in the offspring gradually waned from $\sim 1/4400 \pm 1200$ on day 21 to $\sim 1/90 \pm 60$ on day 70 (Fig. 3e). This is in contrast with no decrease in neutralizing antibody titers in adult animals immunized with 3′UTR-Δ10-LAV (Supplementary Fig. 2). After challenging the 70-day-old pups with $10^6$ PFU ZIKV PRVABC59, 32% of the animals were protected against viremia (Fig. 3f). Correlation analysis between $NT_{50}$ titers and viremia development suggested that a minimal $NT_{50}$ of $1/120$ is required to prevent ZIKV infection accompanied by viremia (Fig. 3g).

The third experiment determined the minimal $NT_{50}$ level of maternally transferred immunity that was required to block in utero transmission when the female offspring became pregnant (Supplementary Fig. 9). Six- to seven-week-old female offspring born to maternally vaccinated dams (subcutaneously immunized with $10^3$ FFU 3′UTR-Δ10-LAV at E0.5) were mated with male mice. At E10.5, the pregnant offspring were bled for $NT_{50}$ measurement and subcutaneously challenged with $10^6$ PFU ZIKV PRVABC59. The pregnant offspring were quantified for viremia at E12.5 as well as for organ viral loads at E18.5 (Supplementary Fig. 9). Correlation analysis suggested that different minimal

$NT_{50}$ titers are required to protect different organs against ZIKV infection during pregnancy: $1/200$, $1/290$, $1/290$, and $1/430$ for viremia, maternal brain, spleen, and placenta, respectively (Supplementary Fig. 9b–i). Since IgG is the main trans-placenta antibody[17,18], these results suggest that antibody immunity alone may be sufficient to prevent in utero transmission.

**Neutralizing antibodies for pregnant and non-pregnant mice.** To validate that antibody alone is sufficient to prevent vertical transmission, we performed passive transfer experiments (Fig. 4a). Ten- to twelve-week-old pregnant mice were passively transferred at E9.5 with different amounts of neutralizing antibodies (derived from ZIKV-immunized mice). At E10.5, the pregnant mice were measured for $NT_{50}$ levels and subcutaneously challenged with $10^6$ PFU ZIKV PRVABC59. The animals were assayed for viremia at E12.5 and organ viral loads at E18.5 (Fig. 4a). The passively transferred antibodies at E9.5 led to detectable neutralizing activities at E10.5, and conferred protection against maternal viremia and viral organ loads at E18.5 in a dose-responsive manner (Fig. 4b–i). Correlation analysis showed that distinct minimal $NT_{50}$ titers were required to protect different tissues/organs against viral infection: $\geq 1/100$ for viremia (Fig. 4b, c); $\geq 1/460$ for maternal brain (Fig. 4d, e), spleen (Fig. 4f, g), and placenta (Fig. 4h, i). In addition, a minimal $NT_{50}$ titer of $\geq 1/460$ was required to prevent fetal weight loss during pregnancy infection (Fig. 4j). Notably, pregnant mice transfused with low levels of neutralizing antibodies (i.e., $NT_{50}$ titers $<1/100$) developed higher viral load in maternal spleen than the animals transfused with placebo control (Fig. 4g), possibly due to antibody-mediated enhancement. However, no such enhancement was observed for viremia (Fig. 4c), maternal brain viral load (Fig. 4e), or placenta viral load (Fig. 4i) in the antibody-transfused animals.

The above results prompted us to examine if pregnant and non-pregnant mice require distinct levels of neutralizing antibodies to prevent apparent ZIKV infection. Ten- to twelve-week-old non-pregnant mice were passively transferred with different amounts of neutralizing antibodies (Supplementary Fig. 10a). On the next day, the mice were bled for measuring $NT_{50}$ titers and subcutaneously challenged with $10^6$ PFU ZIKV PRVABC59. Viremia was assayed on day 2 (peak viremia time) and organ viral loads on day 9 (Supplementary Fig. 10a). Correlation analysis suggested that a minimal $NT_{50}$ titer of $\geq 1/130$ was required to prevent non-pregnant mice from viremia and viral loads in brain and spleen (Supplementary Fig. 10b–g). Thus, the minimal $NT_{50}$ titer required to protect pregnant mice ($\geq 1/460$) was about 3.5-fold higher than that required to protect non-pregnant mice ($\geq 1/130$).

**Weaker T cell response to vaccination during pregnancy.** Besides neutralizing antibodies, T cells play an important role in preventing ZIKV infection and disease[19–21]. Thus, we examined T cell response to maternal vaccination and compared the T cell responses between maternal and non-maternal vaccinations. For maternal vaccination, 12-week-old female mice were mated, and subcutaneously immunized with $10^3$ FFU 3′UTR-Δ10-LAV at E0.5. At E18.5, mouse spleens were harvested for T cell analysis by culturing splenocytes, stimulating them with ZIKV or a viral envelope peptide[19], and analysis by an intracellular cytokine staining (ICS) assay and a Bio-Plex immunoassay. For non-maternal vaccination, age-matched non-pregnant female mice were vaccinated and analyzed in the same manner (Fig. 5a). In both maternal and non-maternal groups, 3′UTR-Δ10-LAV elicited robust ZIKV-specific IFN-$\gamma^+$ and IFN-$\gamma^+$TNF-$\alpha^+$ CD4$^+$ and CD8$^+$ T cell responses (Fig. 5b–d). Consistently, the

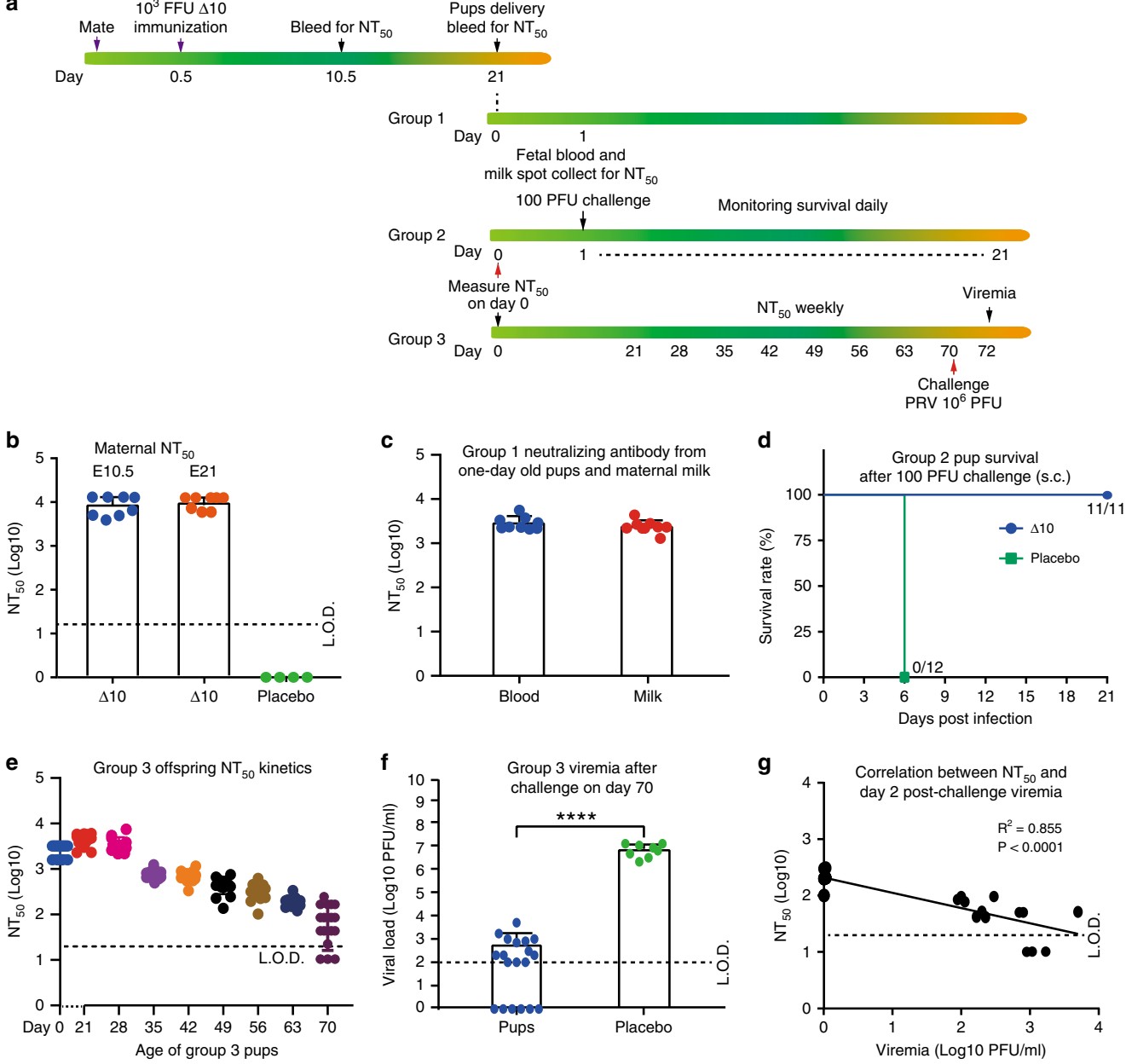

**Fig. 3 Protection of offspring born to dams immunized with 3′UTR-Δ10-LAV at E0.5. a** Experimental scheme. Ten- to twelve-week-old female A129 mice were subcutaneously immunized with $10^3$ FFU 3′UTR-Δ10-LAV (Δ10, $n = 8$) or PBS ($n = 4$) at E0.5. At E10.5, the pregnant mice were bled for measuring neutralizing antibody titers. The pups were delivered after a full-term pregnancy at E21. Maternal and fetal blood was harvested on the day of delivery for neutralizing antibody assay. The neonates were divided into three groups. Group 1 pups were collected for fetal blood and milk spots at one-day-old and measured for neutralizing antibody titers. Group 2 pups were subcutaneously challenged at one-day-old with 100 PFU ZIKV PRVABC59 and monitored for survival for 21 days. Group 3 pups were bled every week after weaning to monitor the decline of neutralizing antibody titers; on day 70, these mice were subcutaneously challenged with $10^6$ FFU ZIKV PRVABC59 and measured for viremia on day 72. **b** Neutralizing antibody titers from PBS- or Δ10-immunized mice at E10.5 and E21. **c** Neutralizing antibody titers from fetal blood and maternal milk collected from one-day-old neonates (Group 1, $n = 9$). **d** Survival of pups after challenge with 100 FFU ZIKV PRVABC59 (Group 2, $n = 11$ or 12). **e** Decay kinetics of neutralizing antibody titers from pups (Group 3, $n = 14$–19). **f** Viremia on day 72 from Group 3 pups. Statistics was performed using Mann-Whitney test. ****$p < 0.0001$. **g** Correlation analysis between day-72 viremia and day-70 $NT_{50}$ values from Group 3 pups. $P$ and $R^2$ values reflect significance and the correlation coefficient. Error bars represent standard deviations. Source data are provided as a Source Data file.

splenocytes from 3′UTR-Δ10-LAV-immunized animals produced significantly higher levels of interleukin-2 (IL-2) and IFN-γ than those from the mock-vaccinated animals (Fig. 5e, f). Notably, compared with non-pregnant vaccination, maternal vaccination elicited significantly lower numbers of IFN-γ⁺ CD4⁺ T cells, IFN-γ⁺ CD8⁺ T cells, and IFN-γ⁺TNF-α⁺ CD8⁺ T cells. It also induced lower levels of IL2 and IFN-γ production (Fig. 5e, f). The

results indicate that pregnancy has a lower T cell response to 3′UTR-Δ10-LAV vaccination.

## Discussion
The feasibility of a safe and efficacious ZIKV vaccine is supported by the availability of clinically licensed vaccines for four

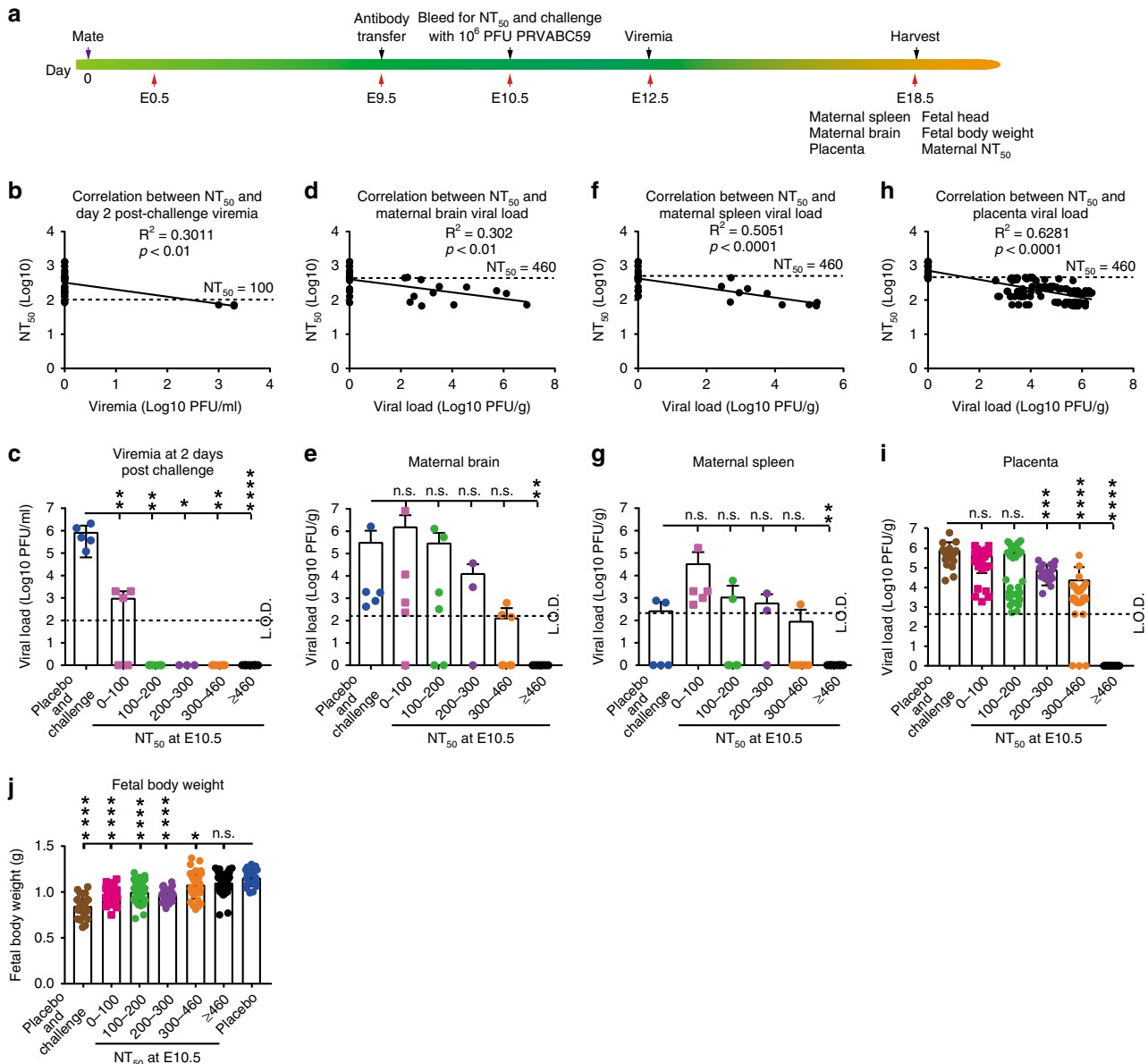

**Fig. 4 The minimal neutralizing antibody titers (NT50) required for protection of pregnant mice. a** Experimental scheme for passive antibody transfer and efficacy test. Ten- to twelve-week-old A129 female mice ($n = 28$) were passively transferred with various amounts of neutralizing antibodies at E9.5. The NT50 titers in recipient mice were determined at E10.5. The pregnant mice were then subcutaneously challenged with $10^6$ PFU ZIKV PRVABC59 at E10.5 and measured for viremia at E12.5. The maternal and fetal tissues/organs were harvested at E18.5 and measured for viral loads. **b** Correlation analysis between NT50 titers at E10.5 and viremia at E12.5. **c** Plot of viremia levels versus different groups of NT50 titers from (**b**). Correlation analyses between organ viral loads (detected at E18.5) and NT50 titers (measured at E10.5) are presented for maternal brain (**d**, **e**), spleen (**f**, **g**), and placenta (**h**, **i**). The dotted lines indicate the minimal NT50 titers required for protection of tissues/organs against viral infection. **j** Correlation analysis between fetal body weight (detected at E18.5) and NT50 titers (measured at E10.5). $P$ and $R^2$ values reflect significance and the correlation coefficient. Mann-Whitney test was performed to indicate significant differences (**c**, **e**, **g**, **i**, **j**). *$p < 0.5$, **$p < 0.01$, ***$p < 0.001$, ****$p < 0.0001$, non-significant (n.s.) $p > 0.5$. Error bars represent standard deviations. Source data are provided as a Source Data file.

flaviviruses (yellow fever, tick-borne encephalitis, Japanese encephalitis, and dengue). Compared with the licensed flavivirus vaccines, a successful ZIKV vaccine has to overcome the unique challenge of protecting pregnant women against in utero viral transmission. Thus, it is essential to determine the correlations of protection against ZIKV infection in non-pregnant individuals as well as against in utero transmission in pregnant women. Given the decline of ZIKV human cases, identifying these parameters has become even more critical to guide the ongoing clinical trials[22]. Our study has demonstrated that neutralizing antibody alone is sufficient to protect against apparent ZIKV infection as

well as in utero transmission in mice. However, a higher neutralizing antibody titer is required to prevent in utero transmission (NT50 ≥ 1/460) than that required to prevent apparent ZIKV infection in non-pregnant mice (NT50 ≥ 1/130). The requirement of NT50 ≥ 1/460 for prevention of in utero transmission was supported by two different types of experiments: pregnant mice with waned antibody levels (Supplementary Fig. 9) and pregnant mice with passively transferred antibodies (Fig. 4). The requirement of NT50 ≥ 1/130 for prevention of apparent ZIKV infection in non-pregnant mice is in agreement with previous studies[10,23–26]. Besides the different requirements

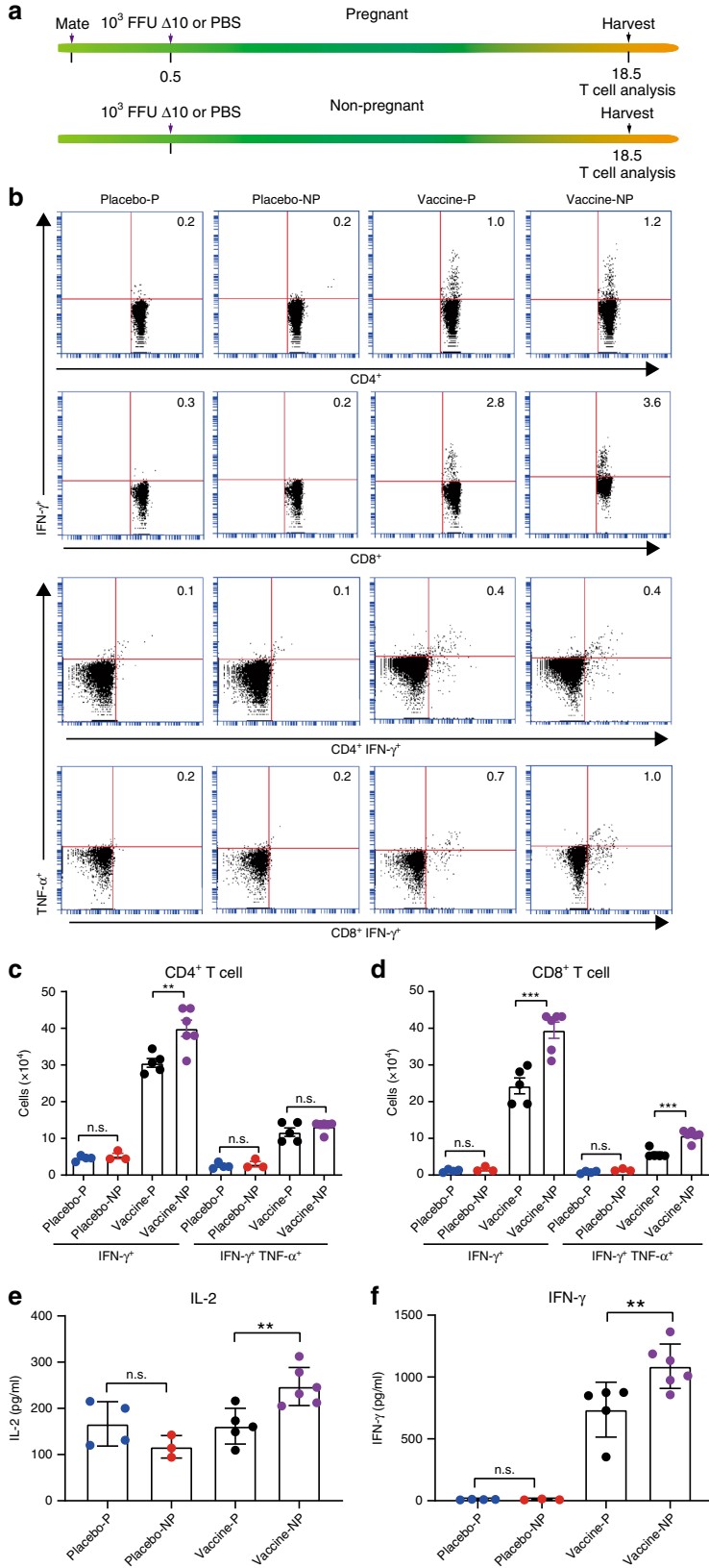

for protective $NT_{50}$ titers between pregnancy and non-pregnancy, we also found that pregnancy has a lower T cell response and cytokine production after 3′UTR-Δ10-LAV vaccination. This is not surprising as pregnancy has been well documented to alter systemic immunity[27–29]. Future studies are needed to define

the mechanism and impact of such pregnancy-mediated T cell attenuation on the development of humoral immunity.

Besides prevention of in utero transmission, ZIKV vaccine development also has to overcome the challenge of antibody-dependent enhancement (ADE) of closely related dengue virus

**Fig. 5 Pregnancy affects T cell response to 3′UTR-Δ10-LAV vaccination in A129 mice. a** Experiment scheme. Top scheme: 12-week-old female A129 mice (n = 4 or 5) were mated with male mice; at E0.5, the pregnant mice were subcutaneously immunized with $10^3$ FFU 3′UTR-Δ10-LAV (Δ10) or PBS; at E18.5, they were sacrificed and splenocytes were harvested for T cell analysis. Bottom scheme: age-matched non-pregnant female mice (n = 3 or 6) were immunized and analyzed for T cell response to Δ10 vaccination, as described above for the pregnant group. **b** Percentages of CD4+IFN-γ+, CD8+IFN-γ+, CD4+IFN-γ+TNF-α+, and CD8+IFN-γ+TNF-α+ cells. Spleen cells were harvested and stained for IFN-γ+, TNF-α+, CD3+, and CD4+ or CD8+. Samples were acquired with BD Accuri C6 Flow Cytometer instrument. Total splenocytes were first gated to exclude, debris, cell fragments, and dead cells based on forward (FSC) and side scatter (SSC). CD3+CD4+ or CD3+CD8+ T cells were next gated for analysis of cytokine production. **c** Total numbers of CD4+ T cell subsets per spleen. Splenocytes were stimulated by ZIKV. **d** Total numbers of CD8+ T cell subsets. Splenocytes were stimulated by a ZIKV E peptide. Cytokines IL-2 (**e**) and IFN-γ (**f**) in cell culture media were measured after splenocytes had been stimulated by the viral E peptide for 3 days. An unpaired t test was performed to analyze statistical significance between indicated groups. **p < 0.01, ***p < 0.001, non-significant (n.s.) p > 0.5. Triple technical replicates were performed for (**c**), (**d**), (**e**) and (**f**). Placebo-P PBS placebo pregnant mice, placebo-NP PBS placebo non-pregnant mice, vaccine-P vaccinated pregnant mice, vaccine-NP vaccinated non-pregnant mice. Error bars represent standard deviations. Source data are provided as a Source Data file.

and vice versa[30–33]. Recent studies reported that passively transferred DENV-specific antibodies or previous DENV infection could increase ZIKV pathogenesis in pregnant mice through increasing damage to the placenta and developing fetus[34–36]. However, some epidemiologic data do not support the presence of immune enhancement in naturally infected humans[37,38]. More epidemiologic studies are needed to further determine if pre-immunity derived from natural DENV or other flavivirus infection enhances ZIKV infection and congenital syndromes[39]. Vaccination with 3′UTR-Δ10-LAV may mitigate the risk of ADE because it can elicit both humoral and cellular immunity against viral structural and non-structural proteins[40]. This concept has been proposed to explain the undesirable outcome for the licensed Dengvaxia[41].

We explored the possibility of maternal vaccination to prevent in utero transmission during pregnancy. Our results indicate that 3′UTR-Δ10-LAV has an excellent safety profile for maternal vaccination: no detectable infectious virus or viral RNA in fetal heads after vaccination, and no detectable adverse effects on murine pregnancy, fetal development, or offspring behavior. Despite these promising results, we could not exclude the possibility that maternal vaccination may cause subtle adverse effect on fetal development that could not be detected by the current assays. Infection of placenta with WT ZIKV was recently shown to mediate pregnancy complications[42,43], suggesting that even a very low level of 3′UTR-Δ10-LAV replication in maternal organs might cause potential immunopathology in developing fetuses. Nevertheless, we showed that a single maternal vaccination of $10^3$ FFU 3′UTR-Δ10-LAV probably conferred sterilizing immunity, prevented apparent in utero transmission, transferred maternal antibodies to pups, and protected neonates against ZIKV infection. However, the protection of offspring declined as the maternal neutralizing antibody titers waned. Altogether, the results suggest that 3′UTR-Δ10-LAV may be considered for maternal vaccination. In support of this idea, maternal vaccination is recommended by the WHO for yellow fever and Japanese encephalitis vaccines during outbreaks in endemic regions or when the risk of viral infection is high[44,45]. Other successful maternal vaccines, such as Tetanus-Diphtheria-Pertussis (Tdap) vaccine and inactivated influenza vaccine (IIV), are recommended by the Center for Disease Control and Prevention (CDC) to all pregnant women[46]. After all, it should be emphasized that an ideal vaccination strategy for ZIKV is to immunize children before they reach child-bearing age.

The current study took the advantage of rapid immune response to 3′UTR-Δ10-LAV immunization to study maternal vaccination and protective immunity for vertical transmission in a ZIKV pregnant mouse model. Compared with LAVs, other vaccine platforms (particularly inactivated ZIKV vaccine or DNA subunit vaccine) may be more appropriate for immunocompromised individuals and pregnant women because of their non-infective nature. Although both inactivated ZIKV vaccine or DNA subunit vaccine require multiple doses, the gestation period of human pregnancy is long enough for using them for maternal vaccination. Considering the strengths and weaknesses of different vaccine technologies, it is important to develop multiple vaccine platforms in parallel to provide complementary options for preventing ZIKV infection and disease. Finally, caution should be taken when extrapolating the mouse results to humans because placental biology and antibody transfer are fundamentally different between mouse and human pregnancy.

## Methods

**Ethics statement.** Mouse studies were performed in accordance with the recommendations in the Guide for the Care and Use of Laboratory Animals of the University of Texas Medical Branch (UTMB). The protocols were approved by the Institutional Animal Care and Use Committee (IACUC) at UTMB (Protocol Numbers 1708051 and 1412070). Subcutaneous and intraperitoneal injections were performed under anesthesia that was induced and maintained with isoflurane. All efforts were made to minimize animal suffering.

**Viruses, antibodies, and cells.** The ZIKV Cambodian strain FSS13025 (GenBank number KU955593.1, referred as WT in text) was produced from an infectious cDNA clone[47]. The ZIKV 3′UTR-Δ10-LAV strain was generated from the FSS13025 infectious cDNA clone[12]. The Zika Puerto Rico strain PRVABC59 (GenBank number KU501215) used for challenge was obtained originally from Dr. Robert Tesh from the World Reference Center of Emerging Virus and Arboviruses (WRCEVA) at UTMB. ZIKV-specific HMAF (hyper-immune ascitic fluid, dilution 1:2,000) was also obtained from Dr. Robert Tesh from WRCEVA at UTMB.

Anti-mouse IgG antibody labeled with horseradish peroxidase (KPL, Gaithersburg, MD) was purchased from SeraCare (Cat. Nos,: 5220-0286, dilution 1:2,000). Vero cells were purchased from the American Type Culture Collection (ATCC CCL-81; Bethesda, MD), and maintained at 37C with 5% $CO_2$ in high glucose Dulbecco modified Eagle medium (DMEM; Invitrogen, Carlsbad, CA) with 10% fetal bovine serum (FBS; HyClone Laboratories, Logan, UT) and 1% penicillin/streptomycin (Invitrogen, Carlsbad, CA). The cell line was tested negative for mycoplasma.

**Quantification of viral load in sera and organs from A129 mice.** Viremia was quantified by plaque assay as described previously[47]. Briefly, Serum was prepared after coagulation and centrifugation. Then 15 μl of sera was used for 10-fold serial dilutions. For each dilution, 100 μl of diluent was added to 24-well plate with 90% confluency Vero cells for infection of 1 h with swirling every 15 min. After 1 h incubation, 0.5 ml of overlay containing 0.8% methyl cellulose, 2% FBS, and 1% Penicillin–Streptomycin were added to each well. The plates were incubated 4 days at 37 °C with 5% $CO_2$ followed with fixing and staining. The maternal and fetal tissues (harvested at E18.5 from pregnant A129 mice) were homogenized by Qiagen TissueLyser II. The supernatant was used for plaque assay to measure the viral load in tissues after centrifuge. To quantify the 3′UTR-Δ10-LAV virus from tissues or blood, immunostaining was performed as previous report[12]. Briefly, after 4 days of incubation, the methyl cellulose overlay was removed, 0.5 ml methanol-acetone (1:1) fixation solution was added per well, and the plate was incubated for 15 min at room temperature. After removing the fixation solution, the plates were washed three times with PBS, incubated in PBS with 3% FBS for 1 h, and reacted with ZIKV-specific HMAF for 1 h. The plates were washed three times with PBS, incubated for 1 h with the horseradish peroxidase-conjugated secondary antibody (KPL, Gaithersburg, MD), and detected for immunostaining by addition of amino

ethylcarbazole substrate following the manufacturer's instructions (ENZO Life sciences, Farmingdale, MA).

**Antibody response from A129 female mice.** Ten-week-old female A129 were immunized with $10^3$ FFU ZIKV-3′UTR-Δ10-LAV by subcutaneous route. The blood was collected by retro orbital sinus (R.O.) bleeding on days 6, 10, and 14 post-immunization. The sera were collected and used to measure the neutralizing antibody titer using an mCherry ZIKV reporter virus[11]. Supplementary Fig. 11 depicts the construction of the cDNA clone of mCherry ZIKV. Briefly, sera were heat-inactivated at 56 °C for 30 min, then sera were 2-fold serially diluted starting at 1:50 in DMEM containing 2% FBS and 1% penicillin/streptomycin and then incubated with an equal volume of mCherry ZIKV reporter virus at 37 °C for 1 h. Antibody-virus complexes were added to Vero cell monolayers in a 96-well plate which was seeded one day before. At 48 h post-infection (p.i.), mCherry fluorescence-positive cells were quantified by Cytation 5 Cell Imaging Multi-Mode Reader (Biotek). The fluorescence-positive cells from serum-treated wells were normalized to those of non-treatment controls (set as 100%). A dose-response curve was plotted using GraphPad Prism 7 software (La Jolla, CA), and the effective dilution of sera to reduce 50% of mCherry-positive cells ($NT_{50}$) was calculated using nonlinear regression analysis.

**A129 mouse pregnancy safety experiment.** A129 mice were bred and housed in pathogen-free mouse facilities at UTMB. Ten- to twelve-week-old female A129 mice were paired with ten-week-old male mice. The vaginal plug was checked daily. Once the plug was identified, the mice were defined at starting embryonic stage (E0.5). The mice were infected with $10^5$ FFU WT ZIKV or 3′UTR-Δ10-LAV by subcutaneous route at E0.5 or E10.5. At E18.5, mice were euthanized and necropsied. Maternal blood was obtained by cardiac puncture. Maternal spleen, brain, and placenta were harvest for measuring viral loads. Fetal body weight was measured immediately after necropsy. After decapitation, fetal head and blood were collected for viral load by plaque assay and neutralizing antibody measurement, respectively. All sera were heat-inactivated at 56 °C for 30 min prior to measuring neutralizing antibodies by the mCherry ZIKV reporter virus as described above.

**A129 mouse pregnancy efficacy experiment.** Ten- to twelve-week-old female A129 mice were paired with ten-week-old male mice. The pregnant mice were immunized with $10^3$ FFU of 3′UTR-Δ10-LAV (Δ10) by subcutaneous route at E0.5 or E4.5. At E10.5, the mice were bled by retro orbital sinus (R.O.), and immediately challenged with $10^6$ PFU ZIKV PRVABC59 via subcutaneous route. At E12.5, mice were anesthetized and bled via the R.O. for viremia testing. At E18.5, mice were euthanized and necropsied. Maternal blood was obtained by cardiac puncture. Maternal spleen, brain, and placenta were harvest for measuring viral loads by plaque assay as described above. Fetal body weight was measured immediately after necropsy. After decapitation, fetal head and blood were collected for viral load by plaque assay and neutralizing antibody measurement as mentioned above, respectively.

**Neutralizing antibody from fetal blood and maternal milk.** The blood was collected from decapitated fetuses at E18.5 or from new born pups or one-day-old pups. The maternal milk was harvested from one-day-old pup's milk spot. After centrifugation of the harvested milk at 12,000 rpm, the supernatant was collected and inactivated at 56 °C for 30 min prior to neutralizing antibody test. All antibody neutralization titers were determined using an mCherry ZIKV as described above.

**Behavioral assays.** Ten- to twelve-week-old A129 mice were immunized with $10^3$ FFU 3′UTR-Δ10-LAV or PBS (placebo group) at E0.5. All the pregnant mice gave birth at full term. The following behavioral experiments were performed on 6-week-old mice. (i) Rotarod test: The motor ability of mice was assessed by determining their performance on a rotarod (AccuScan EzRod, Omnitech Electronics, Inc., Columbus, OH). After three days of training, mice were placed on a rotarod in a ramp acceleration mode (20 rpm/min), and the latency to fall was measured. The values obtained from three trials were averaged on each test day. (ii) Grip strength test: this test was used as another measurement of motor functions. Mice were individually placed on a metal grid attached to a force transducer in a grip strength meter (GT3, Bioseb, Pinellas Park, FL). The mouse was pulled until its grip onto the metal grid is released, and the maximum grip force developed during this procedure was recorded. The values obtained from two trials were averaged. (iii) Tail pressure test: the mechanical nociceptive sensory function of mice was assessed by using a custom-made Randall Selitto device. The tail base of mouse was pressed until the mouse withdraws its tail, and the force evoking such withdrawal was recorded.

**Virulence of ZIKV in one-day-old A129 pups.** To determine the lethal dose of ZIKV PRVABC59 on one-day-old A129 pups, pups were injected with 1, 10, or 100 PFU ZIKV PRVABC59 via subcutaneous route and monitored for morbidity and mortality. To test the maternal antibody protection against ZIKV infection,

one-day-old pups from 3′UTR-Δ10-LAV- or PBS-immunized mice were inoculated subcutaneously with 100 PFU ZIKV PRVABC59. The infected pups were monitored for morbidity and mortality for 21 days.

**Passive transfer of antibody to non-pregnant A129 mice.** Mouse immune sera from 3′UTR-Δ10-LAV immunized A129 mice were pooled and passively transferred into 12-week-old female A129 mice via the intravenous (I.V.) route. One day after serum transfer, the mice were bled and immediately challenged with $10^6$ PFU ZIKV PRVABC59 via subcutaneous route. On day 2 post-challenge, the mice were bled for measuring viremia. All mice were sacrificed on day 8 post-challenge and organs were harvested for measuring viral loads by plaque assay as described above.

**Passive transfer antibody to pregnant A129 mice.** Ten- to twelve-week-old female mice were paired with 12-week-old male mice. Mouse immune sera from 3′UTR-Δ10-LAV-immunized A129 mice were pooled and passively transferred into pregnant A129 mouse via I.V. route at E9.5. At E10.5, the mice were bled and immediately challenged with $10^6$ PFU ZIKV PRVABC59 via subcutaneous route. The mice were bled for measuring viremia at E12.5. At E18.5, dams were euthanized and blood was collected by cardiac puncture. Maternal organs (brain, spleen, and placenta) and fetuses were harvested. Fetal weight was measured immediately after necropsy. After decapitation, fetal head and blood were collected. The viral loads in maternal brain, spleen, and placenta and fetal head were determined by plaque assay on Vero cells[48] as mentioned above. Neutralization antibodies in fetal blood were measured using mCherry ZIKV reporter virus infection assay as described above.

**Intracellular cytokine staining (ICS).** The pregnant mice were immunized with $10^3$ FFU of 3′UTR-Δ10-LAV or PBS by subcutaneous route at E0.5. Age-matched non-pregnant mice were immunized at the same time to serve as controls. Eighteen days after immunization, the mice were sacrificed and splenocytes were harvested for analyses. Approximately $2.5 \times 10^6$ splenocytes were stimulated with $10^5$ FFU live ZIKV (strain FSS13025) for 24 h or 10 mg/mL E peptide (amino acids 294–302 from ZIKV polyprotein) for 5 h. Live ZIKV was used as a stimulant to measure CD4+ T cell response[49]. The E peptide was used as a stimulant to measure CD8+ T cell response[19]. During the final 5 h of stimulation, BD Golgi-Plug (BD Bioscience, Cat. no.: 555029; dilution 1:1,000) was added to block protein transport. Cells were stained with antibodies for CD3 (APC-conjugated; Invitrogen, Cat. no.: 17003182; dilution 1:10), CD4 (FITC-conjugated; BD Biosciences, Cat. no.: 557307; dilution 1:10), or CD8 (FITC-conjugated; BD Biosciences, Cat. no.: 11008182; dilution 1:10). Afterward, cells were fixed in 2% paraformaldehyde and permeabilized with 0.5% saponin. Cells were then incubated with PE-conjugated anti-IFN-γ (Invitrogen, Cat. no.: 12731182; dilution: 1:10) and PE-Cy7-conjugated anti-TNF-α (Invitrogen, Cat. no.: 25732180; dilution: 1:10) antibodies or control PE-conjugated rat IgG1. Samples were processed with a BD Accuri C6 Flow Cytometer instrument. Dead cells were excluded on the basis of forward and side light scatter. Data were analyzed with a Cflow Plus Flow Cytometer (BD Biosciences).

**Bio-Plex immunoassay.** Approximately $3 \times 10^5$ splenocytes were plated in 96-well plates and stimulated with $1.2 \times 10^4$ FFU ZIKV FSS13025 or 10 mg/mL E peptide[19] for 3 days. Culture supernatants were harvested and frozen at −80 °C. Cytokines IL-2 and IFN-γ in the culture supernatants were measured using a Bio-Plex Pro Mouse Cytokine Assay (Bio-Rad) according to the manufacturer's instructions.

**Data analysis.** All data were analyzed with GraphPad Prism v7.02 software. Data are expressed as the mean ± standard deviation (SD). Comparisons of groups were performed using Mann–Whitney test or one-way ANOVA with a multiple comparisons correction. A $p$ value of <0.05 indicates statistically significant.

**Reporting summary.** Further information on research design is available in the Nature Research Reporting Summary linked to this article.

## Data availability
The authors declare that the data supporting the findings of this study are available within the article and its Supplementary Information files, or are available from the authors upon request. The source data underlying Figures and Supplementary Figures are provided as a Source Data file.

## Code availability
There in no custom code generated in this study.

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

## Acknowledgements

The authors thank all members of the Shi lab and colleagues at University of Texas Medical Branch (UTMB) for helpful discussions during the course of this study. The authors thank Dr. Robert Tesh at the World Reference Center of Emerging Virus and Arboviruses (WRCEVA) from UTMB for providing ZIKV PRVABC59 strain and ZIKV-specific hyper-immune ascitic fluid. P.-Y.S. lab was supported by NIH grants U19AI142759, AI127744, Clinical and Translational Science Award UL1TR001439, the Kleberg Foundation Award, John S. Dunn Foundation, Amon G. Carter Foundation, Summerfield Robert Foundation, and CDC grant U01CK000512 for the Western Gulf Center of Excellence for Vector-Borne Diseases. This research was also supported by the World Reference Center for Emerging Viruses and Arboviruses, NIH grant AI120942 to S.C.W. T.W. was supported by NIH grants R01 AI099123 and R01 AI27744, and a grant from Sealy Institute for Vaccine Sciences at UTMB.

## Author contributions

C.S., X.X., H.L., A.E.M., Y.L., M.W., and J-H.L. performed experiments and data analysis. C.S., X.X., H.L., J-H.L, J.M.C, S.C.W., T.W., and P.-Y.S. designed the experiments and interpreted the results. C.S., X.X., H.L., J-H.L, J.M.C, S.C.W., T.W., and P.-Y.S. wrote the manuscript.

## Competing interests

The authors declare no competing interests.
