## [Peer Review File · Nature Communications]

Editorial Note: This manuscript has been previously reviewed at another journal that is not operating a transparent peer review scheme. This document only contains reviewer comments and rebuttal letters for versions considered at Nature Communications .

Reviewers' Comments:

Reviewer #1:

Remarks to the Author:

This manuscript by Shan et al. evaluates a live attenuated Zika virus (ZIKV) administered during pregnancy, and finds that the vaccine is protective in pregnant mice.

The virological assays are solid, but there is no evaluation of histology or immunopathology. My other concerns were more directly addressed.

The vaccine has previously been described, but was not tested in pregnancy until now. I still think this is a very interesting study. In the response to my questions about safety, the authors argue that the vaccine does not cause observable behavioral abnormalities in pups, and that viral RNA was not detected in the fetus or placenta. These findings do suggest that the vaccine is indeed generally safe. In the revised manuscript, the authors now present a more balanced discussion where they point out that they cannot exclude more subtle damaging effects of the vaccine. I'm satisfied with the revisions. I do not have any other concerns or questions.

Reviewer #3:

Remarks to the Author:

Although this manuscript has gone for at least one round of review and several of the reviewer comments were addressed and there were improvements, the specific limitation that the study lacks any mechanistic information regarding why their immunization induces differential responses in pregnant versus nonpregnant mice has not been addressed. Several key experiments suggested by multiple reviewers were not performed specifically relating to understanding immune activation in the mother during pregnancy in response to the vaccine. The information about protective antibody titers is likely to be specific to the virus challenge dose and mouse model, so it is not a major advance in itself to show a threshold for antibody neutralizing titers is required for protection without providing mechanistic information about maternal vaccine protection that could be further translated.

Some major remaining questions include the following:

-whether infection could have been detectable in fetal animals prior to the E18 time point- which is not sufficient to claim no infection since the infection could have been cleared at that late stage.

-the total number of antigen specific activated cells in vivo rather an ex vivo activation % is important in Fig. 5 because pregnancy is known to change the cellular composition of the spleen and the magnitude of the effect could be similar even if the percentage is reduced. (Flow cytometry gating strategies were also still not provided for any figures and this should be required supplemental information.)

The authors claimed in their rebuttal that the methods for the construct of the mCherry reporter virus were included in another publication, but that citation did not seem to contain a description of the virus. The citation also relied on citing a second previous report, Shan et al Nature Medicine 2017

where the authors said, "The construction of the cDNA clone for mCherry ZIKV will be reported elsewhere." Have the authors really previously published the methods for generating the mCherry reporter virus? This needs to be more explicitly addressed in the current paper.

Reviewer #1 (Remarks to the Author):

This manuscript by Shan et al. evaluates a live attenuated Zika virus (ZIKV) administered during pregnancy, and finds that the vaccine is protective in pregnant mice.

The virological assays are solid, but there is no evaluation of histology or immunopathology. My other concerns were more directly addressed.

The vaccine has previously been described, but was not tested in pregnancy until now. I still think this is a very interesting study. In the response to my questions about safety, the authors argue that the vaccine does not cause observable behavioral abnormalities in pups, and that viral RNA was not detected in the fetus or placenta. These findings do suggest that the vaccine is indeed generally safe. In the revised manuscript, the authors now present a more balanced discussion where they point out that they cannot exclude more subtle damaging effects of the vaccine. I'm satisfied with the revisions. I do not have any other concerns or questions.

Response: We thank the review for the positive comments.

Reviewer #3 (Remarks to the Author):

Although this manuscript has gone for at least one round of review and several of the reviewer comments were addressed and there were improvements, the specific limitation that the study lacks any mechanistic information regarding why their immunization induces differential responses in pregnant versus nonpregnant mice has not been addressed. Several key experiments suggested by multiple reviewers were not performed specifically relating to understanding immune activation in the mother during pregnancy in response to the vaccine. The information about protective antibody titers is likely to be specific to the virus challenge dose and mouse model, so it is not a major advance in itself to show a threshold for antibody neutralizing titers is required for protection without providing mechanistic information about maternal vaccine protection that could be further translated.

Response: We agree with the reviewer that understanding the mechanism of pregnancy on immune response to ZIKV vaccine is important. But this is beyond the scope of the current study. We have included this point as a future direction in Discussion.

Some major remaining questions include the following:

-whether infection could have been detectable in fetal animals prior to the E18 time point- which is not sufficient to claim no infection since the infection could have been cleared at that late stage.

Response: We have performed new experiment to demonstrate no fetal infection after pregnant dams were vaccinated. Specifically, we harvested placentas and fetus/fetal heads at E10.5 and

E14.5 after pregnant mice had been vaccinated at E0.5. The data showed that only 34.6% (9/26) and 7.4% (2/27) of placentas were positive by qRT-PCR at E10.5 and E14.5, respectively. None of the fetus/fetal heads had detectable viral RNA. The new results have been added as Supplementary Figure 3.

-the total number of antigen specific activated cells in vivo rather an ex vivo activation % is important in Fig. 5 because pregnancy is known to change the cellular composition of the spleen and the magnitude of the effect could be similar even if the percentage is reduced. (Flow cytometry gating strategies were also still not provided for any figures and this should be required supplemental information.)

Response: We have added the requested data to Figure 5. The flow cytometry gating strategies have also been added to the figure legend.

The authors claimed in their rebuttal that the methods for the construct of the mCherry reporter virus were included in another publication, but that citation did not seem to contain a description of the virus. The citation also relied on citing a second previous report, Shan et al Nature Medicine 2017 where the authors said, "The construction of the cDNA clone for mCherry ZIKV will be reported elsewhere." Have the authors really previously published the methods for generating the mCherry reporter virus? This needs to be more explicitly addressed in the current paper.

Response: The details of mCherry reporter virus construction have now been added as Supplementary Figure 11.

Reviewers' Comments:

Reviewer #3:

Remarks to the Author:

The additional data satisfies the concerns that were previously raised about the data contained within this manuscript. There is still a lack of mechanistic information in this report which the authors have indicated they intend to save for future publications.

REVIEWERS' COMMENTS:

Reviewer #3 (Remarks to the Author):

The additional data satisfies the concerns that were previously raised about the data contained within this manuscript. There is still a lack of mechanistic information in this report which the authors have indicated they intend to save for future publications.

We thank the reviewer for the approval of the manuscript.